



# Technical note: Common glitch affecting the EC/OC split point determination in the Sunset Thermal-Optical Analyzer and recommendations to reduce its occurrence

Stéphanie Gagné[1], Brett Smith[1], Gregory J. Smallwood[1], and Joel C. Corbin[1]

[1]Metrology Research Centre, National Research Council Canada, Ottawa, K1A 0R6, Canada

*Correspondence to*: Stéphanie Gagné (stephanie.gagne@nrc-cnrc.gc.ca)

**Abstract.** We identified a common and relatively frequent glitch in the light transmittance/reflectance measurement during thermal-optical analysis in the Sunset Laboratory bench top thermal-optical analyzer models. In the instrument studied, the glitch is observed for one in three punch analyses when using the default analysis parameters. The occurrence of this glitch can invalidate the split point and thus the OC and EC fractions and absolute quantities reported. The glitch was observed in data from at least three independent laboratories using different thermal protocols. We describe this glitch as a "discontinuity" (rapid increase or decrease occurring over a few seconds) in the laser transmittance or reflectance which happens relatively frequently and whose behaviour varies in amplitude, timing and direction. We use over 2,200 filter-punch analyses to expose the factors that contribute to the risk of such a discontinuity occurring. We demonstrate that these discontinuities are due to the movement of filter punches within the instrument, and can therefore be minimized by decreasing the blower speed of the instrument and, if possible, by ensuring a tighter fit of the filter punch in its holder (by testing different spoons). The decrease in blower speed has a negligible effect on the measured temperature program during analysis and is the single most effective way to reduce the risk of discontinuity occurrence. We recommend these modifications for all Sunset instruments.

## 1 Introduction

Thermal-Optical Analysis (TOA) is commonly used to determine the Organic Carbon (OC) and Elemental Carbon (EC) content of airborne particles as collected on a quartz filter. This technique is used for atmospheric studies (e.g. Schauer et al., 2003; Chow et al., 2009; Brown et al., 2019), engine emissions studies (e.g. Ålander et al., 2004; Hays et al., 2013) and for occupational health studies (e.g. Birch and Cary, 1996; Noll et al., 2007). The instrument was first developed by Johnson, Huntzicker and Cary and descriptions of the method start appearing in the literature at the beginning of the 1980's (Johnson et al., 1982; Huntzicker et al., 1982). Further developments are reported in the following decades (e.g. Chow et al., 1993). Currently, two commercial versions of this instrument exist: one by Atmoslytic Inc. (USA), and another by Sunset Laboratory Inc. (USA). A large fraction of the thermal-optical analyzers in use around the world are from Sunset Laboratory. Sunset Laboratory offers two types of Thermal-Optical Analyzers (TOA): a bench-top instrument and a semi-continuous (field)


instrument (Turpin et al., 1990). The present work focuses on results from the bench-top instrument produced by Sunset

Laboratory Inc.

The functioning principle of the instrument is as follows (see Birch and Cary, 1996 for more details). A sample of airborne particles is collected on a quartz filter. A punch of predefined area is removed from this filter loaded into a holder (spoon) and inserted in the Thermal-Optical Analyzer. In the first phase of the analysis (see Fig. 1), the sample is exposed to a continuous flow of helium while the temperature is ramped up in steps. The temperatures and timing of the program depend on the protocol

used (e.g. Cavalli et al., 2010). The carbon which evolves from the filter during the helium phase is defined as organic carbon (OC) and undergoes oxidation to $CO_2$ before being converted to $CH_4$ for quantification by a flame ionization detector (FID). Although this OC generally corresponds to volatile organic compounds (VOCs), some fraction of VOCs pyrolyze rather than evaporate during the helium phase. After the highest temperature of the helium phase, the sample is cooled before a mixture of 90% helium and 10% oxygen is introduced and the temperature is ramped up in steps again. The carbon which evolves

during this oxidizing phase is converted to $CH_4$ for quantification and is defined as elemental carbon (EC), after subtraction of the pyrolyzed carbon (PC) that formed during the helium phase. Finally, the sample cell is cooled while a known amount of methane is injected into the instrument to adjust the FID signal for drift. During the two phase transitions (helium-to-oxygen phase and oxygen-to-methane) the sample cell is actively cooled with an internal blower (fan).

Throughout the above-mentioned analysis phases, the transmittance or the reflectance of a laser ($\lambda$=698 nm) through or from

(resp.) the filter is monitored. This laser aims to separate PC from EC by defining a "split point". Before the split point, oxidation is considered to be removing PC and the measured signal is attributed to OC. After the split point, oxygen is considered to be removing EC and the measured signal is attributed to EC. This split-point approach assumes that the optical properties of PC are not significantly different to those of EC, and that PC evolves before EC. Although neither of these assumptions are valid (Yang and Yu, 2002), this method remains widely used in recent atmospheric studies (e.g. Phairuang et

al., 2019; Wang et al., 2019) and measurement standards (NIOSH 5040 (NIOSH, 1999), ASTM D6877-13(2018) (ASTM International, 2018), and SAE ARP6320 (2018)).

Numerically, the split point is defined at the time when the laser transmittance or reflectance signal (for thermal-optical transmittance, TOT and thermal-optical reflectance, TOR, respectively) reach the same value as the mean of the first 10 seconds of the analysis when the sample is at ambient temperature or the transmittance / reflectance maximum in the first

heating phases (known as OC1 and OC2) in instances where there is an increase. In these definitions of the measured split point, the repeatability and reproducibility will obviously depend on the quality of the laser transmittance or reflectance measurements. The quality of these measurements is one of the major limiting factors in TOA uncertainties (Boparai et al., 2008). If the laser transmittance or reflectance had a glitch prior to the split point, the split point would be invalid, and the OC and EC values reported by the instrument would be erroneous.

In this study, we found that such a glitch is commonplace in Sunset Laboratory Thermal-Optical Analyzer instruments in multiple laboratories around the world. This glitch manifests as a "discontinuity" (rapid increase or decrease in the transmittance or reflectance signal occurring over a few seconds) which occurs during the transition between the different



phases, i.e., during the periods of cooling between the helium and helium–oxygen phases, and between the helium-oxygen and methane phases. We also show the frequency of this phenomenon on our own bench-top Sunset Laboratory Model 5L instrument and analyze the influence of different operational factors on the occurrence of these discontinuities. Then, we demonstrate that reducing the blower speed helps to reduce the occurrence of these discontinuities. Finally, we give recommendations for best practice for other users of the Sunset Laboratory instruments.

## 2 Method

The data used in this study were generated from analysis of >2,200 thermograms (graphical data plotting temperature, FID, transmittance, and/or reflectance signal versus time as shown in Fig. 1) using a Sunset Laboratory Model 5L Thermal-Optical Analyzer. The samples, including combustion particulate emission samples, sucrose samples (for calibration of the instrument) and blanks were collected on quartz filters (the vast majority were collected by our lab using pre-fired Tissuquartz$^{TM}$ Pallflex$^{®}$ Filters, while a minority were sampled externally to our lab and the specific quartz filters used are unknown for these samples). Several thermal protocols were used, namely a few variants of the so-called NIOSH thermal protocol (Birch and Cary, 1996), EUSAAR 2 (Cavalli et al., 2010), and some slightly-modified thermal protocols of these two with extended time step lengths (around 8% of the punches analyzed). Spoons are handmade and vary in their specification (see Fig. 2 for a picture of two 1.5 cm2 holder spoons). The data presented in this work was always analyzed using the same 1.5 cm$^2$ and 1.0 cm$^2$ spoon to support the punches in the oven. In some cases, we operated the instrument blower at full speed (defined by a value of 16 in the configuration ".par" file) whereas in other cases we operated the instrument at half speed (8 in the .par file).

All figures presented herein are from analyses from the same instrument. However, we have had access to the raw data from other Sunset Laboratory bench-top instruments, where we have directly identified the same behaviour quantified below. Those instruments are listed in Section 3.1. The Sunset Laboratory instrument routinely applies a temperature-based correction to measured laser transmittance signals, to account for the temperature dependence of the optical properties of the quartz filter, and possibly the glass filter holder (Boparai et al., 2008). This correction is based on the measured signals during the final cooling period, at which point no carbon remains on the filter. This correction is not relevant to the present study. As a consequence, the transmittance and reflectance presented are not corrected and the correction is only discussed in Section 3.5.2.

Figure 3 shows examples of discontinuities in laser transmittance through filters. The panels on the left (a,c,e) are thermograms. We observed both increasing and decreasing discontinuities, as well as "recovering" discontinuities where the signal slowly returned to its expected (extrapolated) value after several seconds. Each of the corresponding panels on the right of Fig. 3 (b, d, f) show the laser transmittance at the top, its time differential below and the set temperature (bottom) around the time of the discontinuity.

In this work, over 2,200 thermograms are automatically analyzed, seeking for those that display discontinuities and those that do not. The method as described below is in terms of transmittance, but is equally applicable to reflectance measurements.





Since discontinuities were only observed after the blower was switched on, we developed an automated metric to determine whether a discontinuity is occurring or not, within the first 13 seconds after the start of the blower. This metric is based on the behaviour of the laser transmittance roughly 10 seconds before and after the average discontinuity point (shown in the top of the three subplots on the right panels):

$$\text{DI} = \Delta \text{t} \cdot \text{std} \left[ \frac{\Delta \text{Laser Trans}}{\Delta t} \right]_{t-10s}^{t+8s} , \qquad (1)$$

where DI is the discontinuity indicator, std is the standard deviation, $\Delta$Laser Trans/$\Delta$t is the rate of change of the laser transmittance as a function of time, $t$ is the mean time at which discontinuities occurred (4-5 seconds after the temperature change command), and $\Delta t$ is the time resolution of the data acquisition (1s). The DI is unitless.

In practice, we evaluated Equation 1 over the last 5 seconds of the helium phase and the first 13 seconds of the helium–oxygen phase. This corresponds to a mean discontinuity time $t$ a few seconds of time after the blower was turned on. This represents
the time period in which we observed the discontinuity behaviour, through manual inspection of the data. Some exceptions were identified outside of this time period, but still within the cooling period. They are not captured by our present analysis. Since these exceptions were uncommon, we did not pursue their quantification nor attempt to calculate DI with a moving window to include all possible discontinuities.

As the laser transmittance often changes as a result of variations in the carbon mass or light-absorbing properties of carbon on
the sample filter, or due to changes in the optical properties of the filter or filter holder (Boparai et al., 2008), the DI in Equation 1 was designed to only capture sudden changes in transmittance that are not associated with the changes in particle or filter intrinsic properties. A simple analysis of the difference in laser transmittance before and after a given time period would therefore not qualify as a metric to assess the occurrence of a discontinuity. The middle subplot shows the time differential of the laser transmittance i.e. the rate of change of transmittance with time. The time differential is constant if the laser
transmittance increase or decreases steadily, but will exhibit a larger variation if the transmittance changes suddenly.

The discontinuity indicator DI is therefore the standard deviation of the time differential over the 18 seconds around the discontinuity point, and represents a smoothed quantification of rapid changes in laser transmittance. This metric has the advantage of being automated and allowed us to detect discontinuities that were difficult to observe with manual inspection of the thermograms (some sharp increases). The DI metric values occur over a continuum and there is no single threshold above
which we have a discontinuity and below which we do not. By inspection of the raw data, we define DI < 10 as generally not indicating a discontinuity, while DI > 15 indicates a discontinuity. The samples for which the discontinuity indicator falls between 10 and 15 are inconclusive, as we found samples with and without transmittance discontinuities.



## 3 Results and discussion

### 3.1 Prevalence of laser transmission discontinuities in Sunset Laboratory bench top thermal-optical analyzers

The transmittance discontinuities presented in Fig. 3 are not limited to the Authors' instrument. In fact, we have observed them in many thermograms for samples analyzed with other Sunset Laboratory instruments in laboratories around the world. As part of an interlaboratory comparison, we observed transmittance discontinuities from all three labs involved (using models 4L, 5L and another model 4L). We also identified evidence of numerous transmittance discontinuities in the literature, for example in Cavalli et al., 2010, Fig. 3; Fig. S3.1b; Panteliadis et al., 2015, Fig. 10 (data labelled "11"), Fig. 12 although the

discontinuities don't appear in the classical place, Fig. S18 (labelled "6" and "7"), Fig. S20 (labelled "6"), Fig. S23 (labelled line "3"); and Aakko-Saska et al., 2018 (Fig. 3a, SI Fig. 2.4 (both panels), SI section 5.3 first subplot).

We suspected discontinuities in the data from even more publications than these, however, the quality of the figures was insufficient for us to be certain. Regardless, the problem discussed in this study is widespread. We have identified at least 6 instruments that occasionally produce transmittance discontinuities in the results from their Sunset Laboratory thermal-optical

analyzers.

### 3.2 Experimental conditions and discontinuity likelihood

Since the feature appears to be the same in all the instruments considered, we assumed that the cause and behaviour is the same in all instruments and used data from our own instrument to investigate the factors that influence the probability of the discontinuity occurring and, from these factors, develop recommendations. When discussing this issue with Sunset Laboratory,

it was proposed that halving the blower speed may reduce the incidence of this problem (Sunset Laboratory Inc., personal communication, 2016). We used data from thermal-optical analysis of 2,237 filter punches (blanks, sucrose punches and particulate samples). We also tested the influence of the punch size and the thermal protocol used, for the NIOSH 5040 and EUSAAR 2 protocols.

Figure 4 shows the normalized distribution of the discontinuity indicator for a number different operating conditions. Fig. 4a,

shows a comparison of the distribution of discontinuity indicator for punches analyzed using a blower at full-speed during the cooling step and for those analyzed with a blower at half-speed. These data demonstrate that punches analyzed with the blower at half-speed had a greater frequency with the discontinuity indicator below 10 and a lower frequency with the discontinuity indicator above 15 (see Table 1). Selecting an operating condition that sets the blower at half-speed during the cooling steps of the thermal protocol reduced the chances of a discontinuity occurring.

Figures 4b and 4c show the normalized distributions of discontinuity indicator comparing 1 $cm^2$ punches and 1.5 $cm^2$ punches using b) a half-speed blower and c) a full-speed blower. In Fig. 4b, using a half-speed blower, the punch size doesn't seem to influence the occurrence of transmittance discontinuities with most of the discontinuity indicator distribution being below 10. In Fig. 4c, however, using a blower at full speed, the discontinuity indicator distributions is skewed towards the right with a significant fraction of the DI counts above 15 for both punch sizes. In this case, the feature may depend on the spoons used to



hold the punch in the oven. We used one spoon of each size and observed that the 1.5 cm$^2$ punches fitted more securely in their spoon than did the 1 cm$^2$ punches. These spoons are hand-made and therefore have a significant variability in their dimensions (Fig. 2). This result may therefore not generalize to all other instruments. Regardless, this result demonstrates that a loosely-fitted filter punch leads to more discontinuities, and it leads to the below-mentioned hypotheses on the cause of these discontinuities. Halving the blower speed, as illustrated in Fig. 4b, appeared to significantly reduce the difference in

discontinuity risk between the two punch sizes.

Figure 4d and 4e show the normalized probability distributions of the discontinuity indicators separated according to TOA protocol (NIOSH 5040 or EUSAAR 2) using either d) a half-speed blower or e) a full-speed blower. In Fig. 4d, while there are more discontinuity indicators above 15 and in the inconclusive zone (10-15) for the NIOSH 5040 protocol than for EUSAAR 2, the vast majority of all samples regardless of thermal protocol were below 10 for samples analyzed with the

blower operating at half-speed. We attribute the larger DIs distribution for DIs < 10 in NIOSH 5040 than in EUSAAR 2 to the larger cooling step in the NIOSH 5040 than in the EUSAAR 2 protocol (-320°C versus -150°C, respectively), which causes the transmittance to change with temperature less steadily, and thus yield a slightly larger discontinuity indicator (but still below 10). Our data suggest that discontinuities are less likely during the EUSAAR 2 protocol (Fig. 4e), although only 16 samples were available to support this conclusion, in contrast to the >200 samples available for all other thermal-dependent

analyses.

The overall conclusion we draw from this analysis is that operating the blower at half-speed during the cooling periods is the single most effective way to reduce the occurrence of a discontinuity, without any apparent drawbacks. The sample appears to cool at the same rate, and reaches the desired temperature in the same time. Although for our instrument, the 1.5 cm$^2$ punch gave better results than the 1.0 cm$^2$ punch, we attribute this difference to the degree to which of our filter-holding spoon secure

the filter-punch (see Fig. 2) rather than to a systemic issue with smaller punches in general. Buying different spoons until a spoon with a secure fit is found could be expensive. Similarly, changing protocols to reduce the risk of discontinuities occurring may render the data difficult to compare to other data in the field. Changing the blower speed seems to be the cheapest and best solution for safeguarding data continuity.

### 3.3 Discontinuities in reflectance

Considering how frequently discontinuities in transmittance are observed for samples analyzed with operating the blower at full-speed (about a third of the samples with discontinuity indicators above 15), consideration was directed towards investigating if the same behaviour is observed for reflectance. Figure 5 shows a clear example of a discontinuity in reflectance around the time where the blower starts. The number of samples with discontinuity indicators above 15 for reflectance also represent about a third of the samples. However, as the reflectance tends to be noisier, especially when the filter is lightly

loaded (incl. sucrose and blank samples), the discontinuity indicator scale used for transmittance cannot necessarily be directly applied to reflectance: the thresholds between the no-discontinuity zone, inconclusive zone, and discontinuity zone are unlikely to be the same. We did, however, observe many reflectance discontinuities in our samples. The normalized discontinuity





indicator distribution (not shown) suggests that the same factors are associated with higher values of the discontinuity indicator i.e. the blower full speed and the 1 cm$^2$ punch size increased the probability of a discontinuity in the reflectance. As with transmittance, the significance of the punch size is not the size, but how securely the spoon holds the punch. Our 1 cm$^2$ spoon does not secure the punches as effectively as our 1.5 cm$^2$ spoon.

**3.4 Discontinuities linked with punch displacement**

Since the operational speed of the blower is the single-most important factor influencing the likelihood of a discontinuity, this leaves us with two hypotheses to explain these discontinuities: 1) the blower turning on results in a power fluctuation in the laser output; and 2) the air movement or vibrations cause by the blower causes the filter to move.

**3.4.1 Hypothesis 1: power fluctuations affecting the laser output power**

In Fig. 6, we present a collection of different transmittance-reflectance discontinuity combinations that were discovered in our sample dataset. If the laser output power was affected by the power demand of the blower, we would expect transmittance and reflectance discontinuities to occur simultaneously, registering most likely a decrease of both (Fig. 6a), and possibly a recovery as the blower is turned off. Although what looks like discontinuity recoveries have been observed in some thermograms, they are not the norm. Fig. 6b shows a sharp increase in reflectance while transmittance is unaffected, indicated with a black arrow, followed by a sharp decrease in transmittance a few seconds later corresponding to a sharp decrease in reflectance, indicated with a blue arrow. If the laser power supply was the problem, one would expect the magnitude of the discontinuities to be increasing or decreasing together. In Fig. 6b, the magnitude of the reflectance discontinuity under the black arrow is larger than under the blue arrow; the opposite is observed for transmittance. Fig. 6c shows an example where there are only discontinuities in the reflectance while the transmittance appears unaffected. Finally, Fig. 6d shows clear discontinuities in different directions: transmittance decreasing sharply and reflectance increasing sharply. Because of cases like those in b, c and d, we rule out the hypothesis that the discontinuities are due to laser power fluctuations related to operation of the blower.

**3.4.2 Hypothesis 2: punches moving during analysis**

Since the discontinuities are related to blower operation, and since power fluctuations of the laser have been ruled out, we consider that a movement of the filter as the air is blowing is the only logical explanation left.

To support the idea that movement of the punch could have the effect observed, we tried measuring the laser transmittance through a blank filter when setting the spoon at different angles from roughly -30° to +30° with 5 repeats at each angle. The results of this experiment are shown in Figure 7 and are consistent with the magnitude of the discontinuities observed in the analyzed dataset. While we do not claim that the spoon itself has to move for a discontinuity to occur, we use this experiment to get an idea of the magnitude by which the laser transmittance would change if the filter were to be displaced or its angle changed during the analysis.





With this evidence in mind, we conclude that the most likely reason for the transmittance and reflectance discontinuities is that punches are moving during the cooling steps, and that this is due to either blower-related vibrations or air flow. Reducing the
speed of the blower to half has reduced the likelihood of discontinuities and has not affected the cooling rate significantly.

### 3.5 The consequences of discontinuities on the determination of the split point

There are two cooling steps in a standard thermal protocol. The first one is in the analysis phase at the transition between helium and helium–oxygen and on which we have focused in this paper; the second one is after the analysis proper, during the methane calibrating phase (shown in Fig. 1). While only the consequences of transmittance discontinuities happening in each
of these cooling steps are discussed below, the same insights are applicable to reflectance as well.

Transmittance discontinuities only affect transmittance-based split points and reflectance discontinuities only affect reflectance-based split points. Transmittance and reflectance based split points can coincide in some samples but they are known not to always give the same result in terms of the timing of the split point and therefore fractions and absolute quantities of the evolved carbon associated with the OC and EC phases (Chow et al., 2004). Due to the potential bias between the results
obtained with transmittance and reflectance modes of operation, one cannot necessarily switch from one mode to the other if only one of them was affected by a discontinuity (as shown in Figure 6b).

### 3.5.1 Consequences of discontinuities occurring during the first cooling step, in the analysis phase

The consequences of a discontinuity on the split point determination and the quantity of OC and EC reported depends on whether the split point occurs before or after the discontinuity. If the split point is before the discontinuity i.e. in the so-called
OC phase, then the discontinuity did not change the split point determination. However, if the split point is after the discontinuity, then the split point is incorrect and the fraction of OC or EC will either be too big or too small depending on the direction of the discontinuity (increasing or decreasing). The magnitude of this error will depend on the magnitude of the discontinuity and the sensitivity of the split point i.e. by how much the transmittance varies in time (ignoring the discontinuity) and the amount of carbon evolving around the split point.

### 3.5.2 Consequences of discontinuities occurring in the cooling step in the methane phase

The consequences of discontinuities occurring during the cooling step at the end of the entire thermal protocol, during the methane phase, affect the temperature correction applied by the manufacturer's software. The temperature correction method can vary depending on the user's selection but all methods apply a fractional correction based on the variation of the transmittance as a function of temperature during the methane phase. If there is a discontinuity during that phase, the correction
applied will be different and may change where the split point falls. In this case, the magnitude of the error caused by the transmittance discontinuity also depends on the magnitude of the discontinuity and the sensitivity of the split point.

For discontinuities of a given magnitude, the effect on the fractions of OC and EC reported will be much smaller if the discontinuity occurs in the methane phase than for discontinuities happening during the analysis phase due to it being a second



order effect. The laser transmittance in that phase is only used for the temperature correction which is normally of the order of
less than 10%. If the discontinuity is small and the laser slope of the decay in temperature during the methane phase appears
roughly unaffected, it is reasonable not to be concerned with the error the discontinuity will introduce. Since the temperature
correction methods are based on the slope of the transmittance during the methane phase and not its absolute values, a
discontinuity during the analysis phase will not influence the temperature correction itself.

## 4 Conclusions and recommendations

In this study on the operation of Sunset Laboratory thermal-optical analyzers, we observed the following:

- Transmittance/reflectance discontinuities are common and were observed to happen on several instruments in multiple laboratories.
- These discontinuities can affect the determination of the split point and, by consequence, the reported OC and EC fractions and absolute quantities.
- 260    The most important factor affecting the frequency of transmittance/reflectance discontinuities is the operation of the blower, where reducing the blower speed to half of the full speed significantly reduced the risk of a discontinuity occurring.
- When using the blower at full speed, the degree to which a spoon holds the filter punch is important.
- The discontinuities are likely caused by movement of the filter punch due to either air flow or vibrations related to 265    the operation of the blower, and most often occur at the start of the blower operation.

Based on these observations and on our experience analyzing punches, we recommend the following:

1. Operate the blower during the cooling steps at half-speed by default. These settings can be changed in the instrument control ".par" file from a value of 16 to a value of 8.
2. Ensure that punches of a given size fit securely into the spoon provided by the manufacturer. Replace spoons when 270    the fit is not secure.
3. Always inspect thermograms for discontinuities as a quality control step. This can be most effectively done with an algorithm such as that presented in Eq. 1, or through manual observation of the thermogram. From our experience, even with a spoon that does not hold the filter punch securely, as long as the blower is operated at half speed, >84% of the results do not demonstrate a discontinuity (DI < 10), and the assignment of the split point should be correct.
4. If a transmittance/reflectance discontinuity occurs during the methane phase, users must make a risk assessment and choose whether to discard the analysis, correct it or use it *as is*.
5. If a transmittance/reflectance discontinuity occurs during the analysis phase and the split point is affected (split point after the discontinuity), it is best to discard the result completely. If there is space on the filter for another sample, remove another punch from the filter and attempt the analysis again.

## 5 Author contributions

S.G. noticed the discontinuity behaviour, analyzed the data and wrote the manuscript; B.S. operated the thermal-optical
analyser, formulated hypotheses; G.J.S formulated hypotheses regarding the cause of the discontinuities and contributed to
writing; J.C.C. contributed to data analysis and writing. All co-authors have reviewed the manuscript.



## 6 Acknowledgements

We gratefully acknowledge funding from Natural Resources Canada through the PERD TR3 Project (3B03.0002B) and NRCan grant PERD-EIP-EU-TR-04A. Sunset Laboratory Inc. is acknowledged for fruitful discussions regarding the discontinuity issue. Drs. Kevin Thomson, Linda Johnston and Prem Lobo are acknowledged for their role as team leaders during the course of this work. Dr. Fengshan Liu, is acknowledge for securing funding through the PERD-TR3 Project.

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

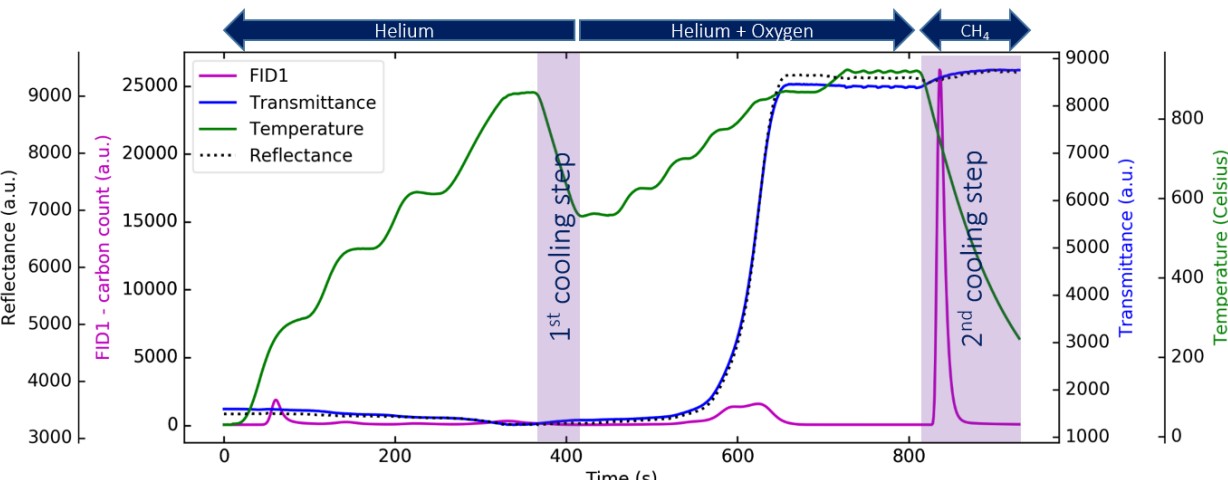

**Figure 1. Example of a typical thermal protocol (in this case the NIOSH 5040 protocol) in the Sunset Laboratory Inc. bench-top**
**Thermal-Optical Analyzer. The filter punch loaded with PM undergoes a step-wise heating process in helium flow until the end of the first cooling step, then undergoes a step-wise heating process in helium-oxygen flow until the start of the second (methane) cooling step whence the temperature correction is derived.**

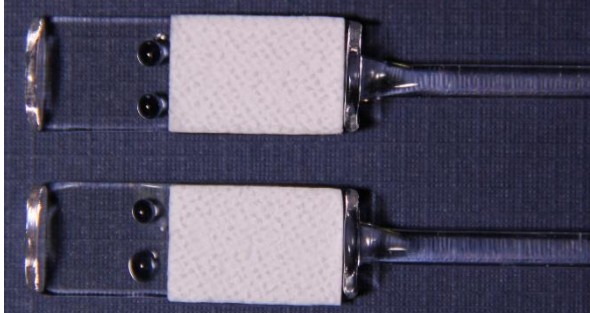

**Figure 2. Two spoons for 1.5cm² punches loaded with punches of equal sizes.**

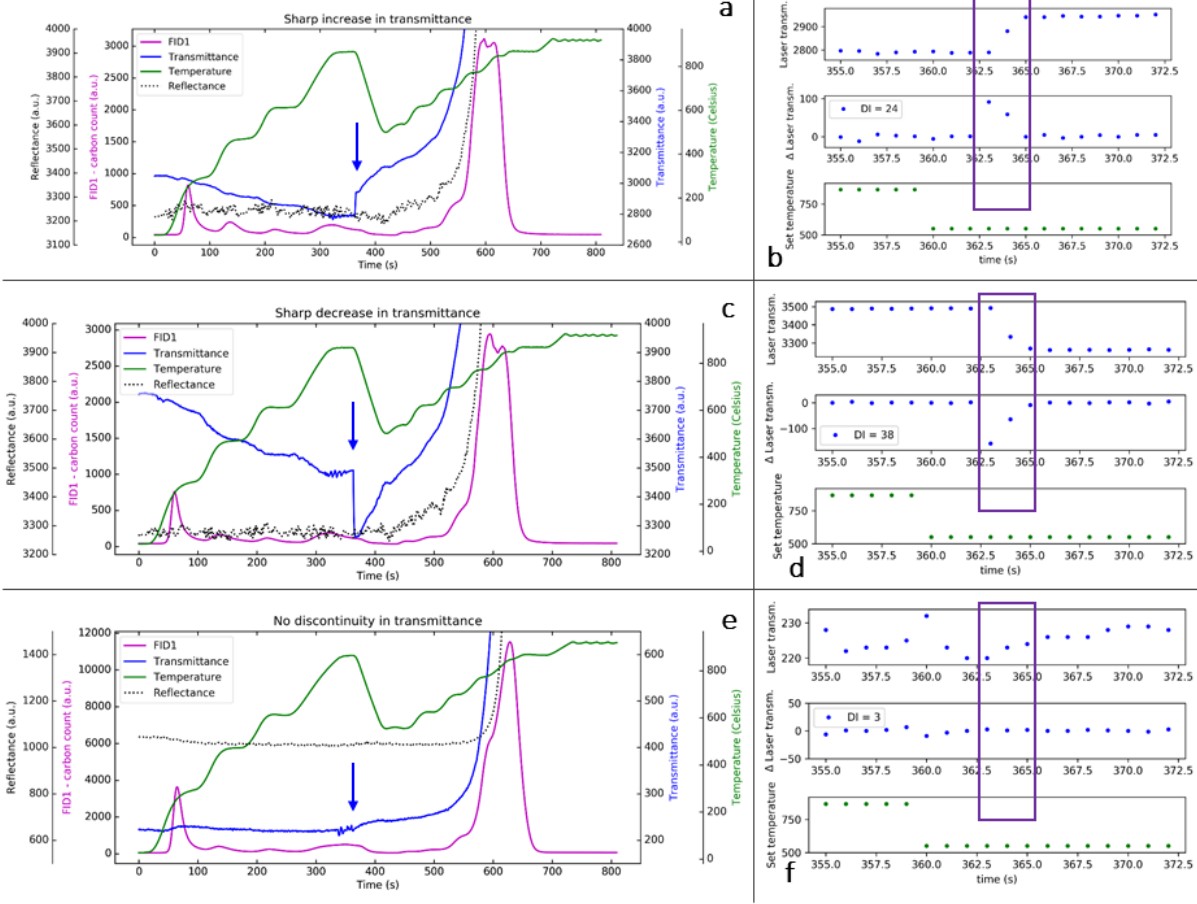


**Figure 3. Examples of transmittance discontinuities increasing sharply (a), decreasing sharply (c), and no discontinuity (e) for three black carbon samples. On the right are the corresponding demonstrations (b, d, and f) of the discontinuity indicator calculation method with the upper panel showing the laser transmittance, the middle panel showing the time differential of the laser transmittance along with its standard deviation around the discontinuity site indicated as *DI* (discontinuity indicator), and the lower**
**panel showing the desired temperature, indicating the timing of the signal to start the blower.**


**Figure 4. An investigation of the factors that may influence the likelihood of a discontinuity in transmittance occurring. Each subplot shows the normalized distributions of discontinuity indicators for different subsets of the data, with the number of analyses indicated as *n* in parentheses in the legend. Subplot a) shows the samples analyzed with variants of the NIOSH 5040 thermal protocol and divided in those analyzed with the blower speed setting at full speed or half speed, as defined in the text. Subplots b) and c) show the distributions of discontinuity indicators at full (c) or half (b) blower speed for data grouped by filter-punch areas (1.5 cm² area or 1.0 cm²) using variants of the NIOSH 5040 thermal protocol. Subplot d) and e) show the distributions of discontinuity indicators for**





**different TOA protocols (variants of NIOSH 5040 or EUSAAR 2) at half (d) or full (e) blower speed regardless of the filter punch size.**

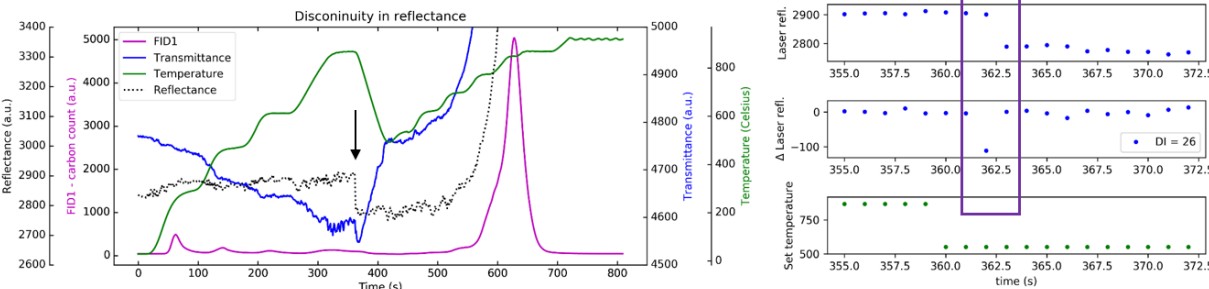


**Figure 5. Example of a discontinuity in reflectance. On the left, a thermogram with the reflectance decreasing sharply shortly after the blower was turned on. On the right, the laser reflectance, time differential of the laser reflectance and set temperature as a function of time. The discontinuity indicator is in the box identified as *DI*.**

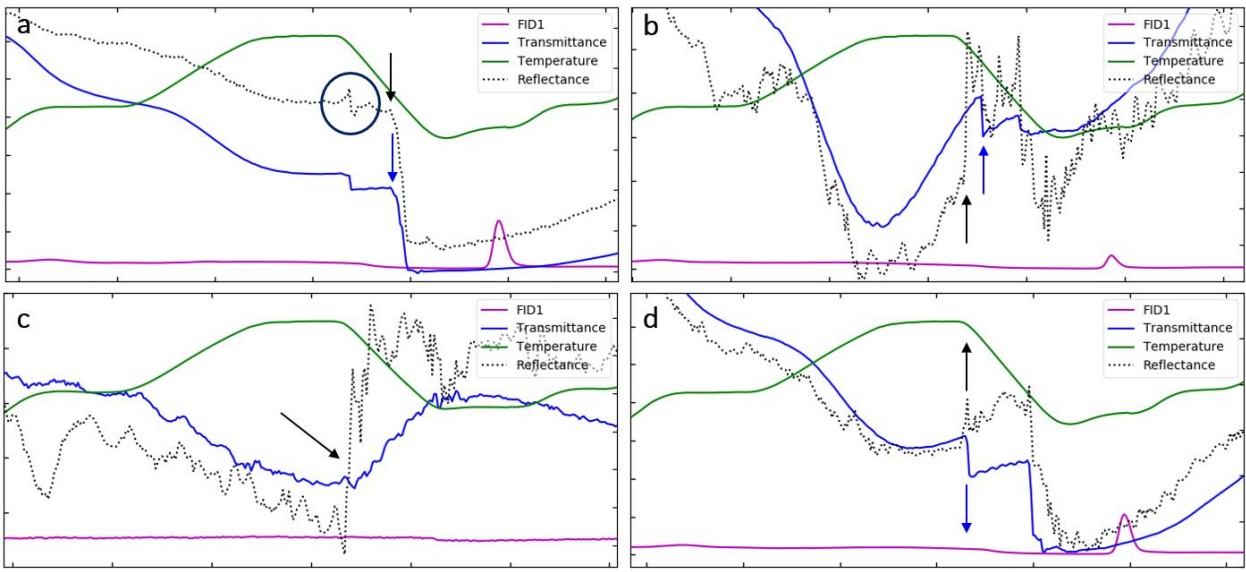


**Figure 6. Examples of transmittance and reflectance discontinuities: a) transmittance and reflectance both decrease sharply and simultaneously; b) transmittance and reflectance discontinuities do not occur simultaneously; c) while reflectance increases sharply, transmittance does not exhibit a discontinuity in a way that is distinguishable from the noise level; d) transmittance decreases sharply while reflectance increases sharply. Note the apparent return to normal reflectance in a), in the circle. This illustrates another discontinuity type briefly discussed in Section 2.**





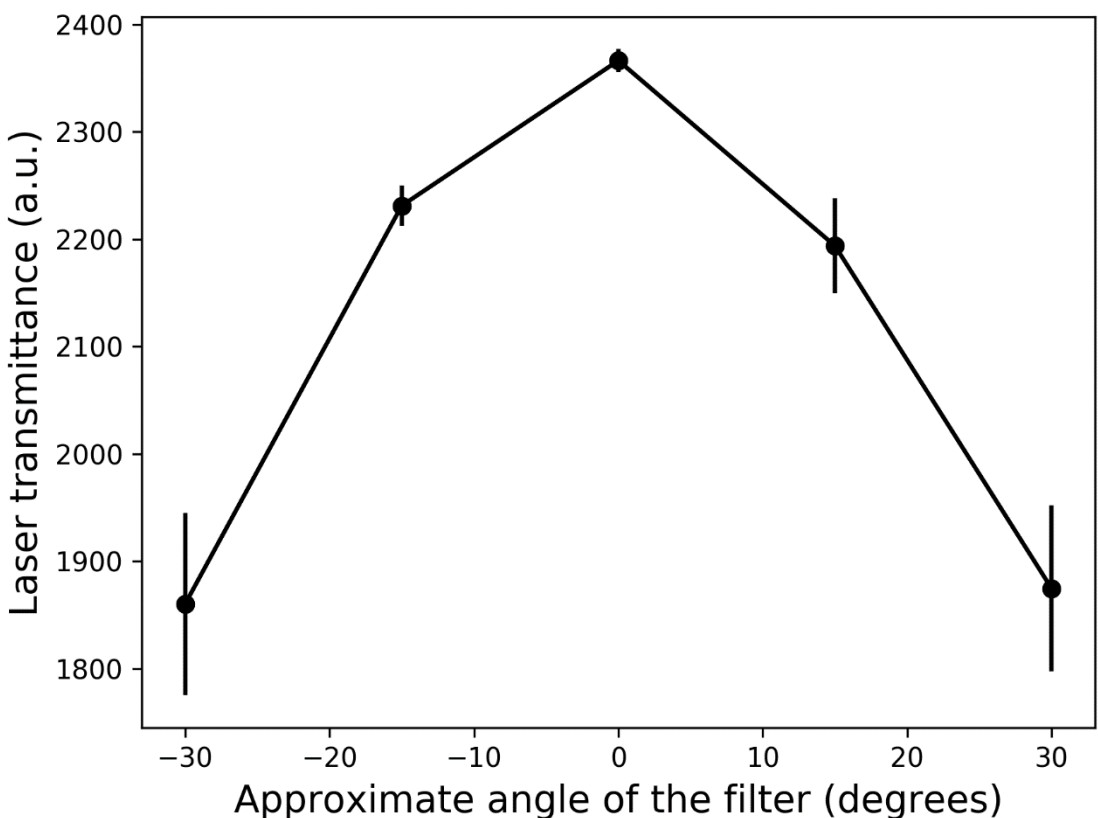


**Figure 7. Laser transmittance through a blank filter with the spoon placed at different approximate angles. The dots represent the mean of the transmittance and the bars represent the standard deviation for 5 repeats at the same approximate angle.**

**Table 1. Fraction of punches with a DI below 10, between 10 and 15, and above 15 for the data presented in Fig.4.**

| | DI < 10 | 10 < DI < 15 | DI > 15 | # samples |
|---|---|---|---|---|
| **a) Blower speed effect (NIOSH only)** | | | | |
| **Blower at full speed** | 43.9% | 23.1% | 33.1% | 859 |
| **Blower at half speed** | 78.8% | 10.8% | 10.4% | 1375 |
| **b) Spoon size effect (blower at half speed, NIOSH only)** | | | | |
| **1.5 cm$^2$ punch** | 86% | 9% | 5% | 87 |
| **1 cm$^2$ punch** | 84.3% | 11.9% | 3.8% | 948 |
| **c) Spoon size effect (blower at full speed, NIOSH only)** | | | | |
| **1.5 cm$^2$ punch** | 48.8% | 19.4% | 31.9% | 480 |





| | | | | |
|---|---|---|---|---|
| **1 cm² punch** | 28.2% | 37.1% | 34.7% | 213 |
| **d) Thermal protocol effect (blower at half speed, all punch sizes)** | | | | |
| **NIOSH 5040** | 78.8% | 10.8% | 10.4% | 1375 |
| **EUSAAR 2** | 93.5% | 4.3% | 2.2% | 230 |
| **e) Thermal protocol effect (blower at full speed, all punch sizes)** | | | | |
| **NIOSH 5040** | 43.9% | 23.1% | 33.1% | 859 |
| **EUSAAR 2** | 81% | 13% | 6% | 16 |
