# Peer review of "Technical note: Common glitch affecting the EC/OC split point determination in the Sunset Thermal-Optical Analyzer and recommendations to reduce its occurrence"

_Atmospheric Measurement Techniques, 2019_

## Referee Comment (RC1) · Anonymous Referee #3 · 23 Sep 2019

The authors of this technical note present a technical issue observed with their analyzer during OC/EC analysis. Based on their findings and respective troubleshooting they recommend similar actions to be applied by other users. The presentation quality of the manuscript is good and very detailed yet the scientific input not as significant. In principle it lies upon the user to regularly verify the stability of the laser signal of their analyzer and include it in the daily standard operating procedures. For example, EN16909:2017 describes a laser stability test carried out during instrument blank analysis throughout which the laser signal should not deviate more than +/- 3% from

its average value. When greater deviations are observed then troubleshooting should follow before proceeding in further analysis. The technical note at its current form gives the feeling that the authors are trying to generalize their instrument specific observation to a universal analyzer behavior and conclude in suggesting preventive actions for all analyzers in use. It is recommended that the technical note goes through major revisions Including alterations related to language and conclusion points moderation prior to final publication. Some of the recommendations for revision are listed below: The term "glitch" does not seem appropriate to describe the authors' observations. The terms "deviation" or "shift" could be alternatively used. Line 18. This sounds like a preventive action even for analyzers that have not shown any similar behavior. On what grounds would that be based? Line 60: "multiple laboratories around the world". It seems that there is a generalization without the evidence to support it. Line 76: Following the text is appears that the "Same", defect spoon was applied further on for analysis. If that is the case, wouldn't the Issue have been avoided with a simple spoon replacement? Line 84: Nonetheless, the laser correction is relevant for the determination of the split point and it is critical for minor alterations of the laser signal during high temperature steps. When marginal, noise related laser deviations appear in raw data then software correction may compensate for the effect. Line 98: It seems that the equation applied does not account for a relative deviation of the laser signal but rather for absolute. If this is the case, then a deviation from 3000 to 3050 counts is accounted equally to a deviation from 300 to 350. Have the authors considered a DI equation that would investigate the discontinuity relatively to the background laser signal, e.g. % of difference compared to background signal average? Further would there be any duplicate analysis on samples where a high DI was observed that could demonstrate the actual shifts of the split points and the differences in reported OC/EC concentrations? Line 118: Nevertheless, a routine "manual" inspection of thermograms could easily help identifying cases like shown in Figure 3a and c, which are eye-catching and lead to respective troubleshooting. Line 126: 3 analyzers are mentioned here, 2 of which are from the 4L series, which

is a different, earlier released model. No information on the optical set-up or the age of the analyzers is provided. No data or graphs presented for all the analyzers mentioned. Could the authors elaborate a bit more on the instrument specifics of this study? Line 132: Would these 6 instruments include the 3 mentioned earlier in the current study and 3 from the literature? Related to the 3 publications referenced for similar observations: • Aaako-Saska et al: Wet punches were analyzed involving additional handling and overall challenging samples. Not representative ambient samples for OC/EC analysis and this is one of the main points the authors are trying to communicate in this paper. • Cavalli et al: Figure 3 of this publications relates to natural granular Calcite analysis. Certainly not representative for OC/EC analysis and probably the laser signal deviations relate to the temperature rather than spoon moves or blower vibrations. • Panteliadis et al: It seems that this study illustrates quite the opposite from what the current technical note communicates across. None of the 17 participants of the comparison exercise showed an occurrence like the one in Figure 3 of the current manuscript. Even though a couple of the participants of this comparison exercise showed a rather poor laser stability and performance none of them was identified to have issues with loose filters on the spoon or blower vibrations. All in all, the vast majority of the thermograms and laser signal graphs in these publications demonstrate quite the opposite from what is shown in this manuscript. There are no remarkable laser signal deviations observed in any of the cases. Further, the authors seem to have only visually observed the graphs included in the referenced papers and not actually investigated any raw or corrected laser data in order to rate them in the suggested DI scale. Figure 3e: The noise of the laser signal seems to be related to the high temperature step and not during blower operation. Similar noise appears in graphs 3a and 3c at an earlier point but during high temperature steps of the Helium mode. Same applies for Figure 5. The axis scale does not allow to check for similar behavior during the high temperature step during the oxidizing phase. Have the authors observed such behavior? This could be also the result of a soiled oven or a filter containing refractory inorganic matter. Have the authors identified such

filter samples or observed a soiled oven when it was eventually replaced? Can such laser behavior affect the calculations of the DI? In Figures 3, 5 and 6 the Reflectance signal seems to be noisy during the whole analysis run indicating other issues than the spoon shape and blower vibration. Could the authors comment on the laser reflectance signal behavior overall? Figure 6: Axis units need to be included. The scale seems to have been magnified compared to the rest of the graphs. 3.5.2 Did the authors investigate and/or observed such discontinuities of the laser signal during the methane phase, in terms of DI numbers? One would expect that these would occur with the same frequency as during the first cooling phase, since at this point the blower operates once again in full speed. Table 1: Large DI was observed in significantly lower frequency for EUSAAR2 compared to NIOSH870. Further, it seems that the laser shifts in the data presented here occur mostly in the first seconds of the blower's operation. Considering that the blower during EUSAAR2 is still working on full speed for 30 seconds, what would the reason be for the protocol-dependent difference? Could it be the case that this issue is related to high temperatures?

Please also note the supplement to this comment:
https://www.atmos-meas-tech-discuss.net/amt-2019-298/amt-2019-298-RC1-supplement.pdf

---

## Referee Comment (RC2) · Anonymous Referee #1 · 2 Oct 2019

General comments The paper describes a "glitch" that is sometimes observed in the laser signal for EC/OC analysis, conducts an extensive investigation of its occurrence and cause, and proposes a simple alteration to the normal operating parameters that reduces the observed numbers of glitches, together with other recommendations. This is a significant issue with a widely-used analysis method, and merits publication on those grounds alone.

Specific comments The metric (the "discontinuity indicator") used to automatically detect the glitch in a large number of samples is somewhat subjective, but nevertheless

fit-for-purpose in evaluating the extent of the issue in different circumstances.

From Table 1, the reduction in blower speed generally decreases the incidence of significant discontinuity (DI > 15) by a factor of about 3. The Abstract should be more quantitative about the effectiveness of halving the blower speed and the need for other QA/QC measures.

On page 6 line 171 it is stated that the change in flow rate does not change the cooling rate. Some evidence needs to be given for this.

Section 3.4.2 illustrates the effect of filter movement on laser transmittance by rotating a filter by 30° from its normal position. This is unconvincing. The paper should estimate the angular movement that a filter held in place by the spoon could realistically experience, and interpret the experimental results in terms of the effect on the discontinuity indicator.

Section 3.5 discusses consequences on the EC/OC split. It would be very helpful to have some examples of the quantitative effect on the EC/OC (or EC/TC) ratio, rather than saying that this depends on various factors.

Figure 4 shows anomalous peaks in the bins just below 50. Is this because higher values are combined?

Technical corrections Page 2, line 32: comma needed after "filter" Line 45 "resp." should be "respectively" Line 48 "and" should be "or" Page 3, line 76 "cm2" should be "cm^2" Page 4, line 113 insert "in Figure 3" after "subplot" Page 5, line 136 "the same" should be "similar" Page 14 Figure 4 labels its x-axes as "jump indicator". This should presumably be "discontinuity indicator" for consistency with the text.